# Autoregressive, Yet Revisable: In Decoding Revision for Secure Code Generation

**Chengran Yang** [1]  **Zichao Wei** [2]  **Heminghao Deng** [2]  **Jinfeng Jiang** [1]  **Zhensu Sun** [1]  **Ting Zhang** [3]  **Tianyi Wu** [4]
**Ming Wen** [2]  **David Lo** [1]

## Abstract

Large Language Model (LLM) based code generation is predominantly formulated as a strictly monotonic process, appending tokens linearly to an immutable prefix. This formulation contrasts to the cognitive process of programming, which is inherently interleaved with forward generation and on-the-fly revision. While prior works attempt to introduce revision via post-hoc agents or external static tools, they either suffer from high latency or fail to leverage the model's intrinsic semantic reasoning. In this paper, we propose Stream of Revision, a paradigm shift that elevates code generation from a monotonic stream to a dynamic, self-correcting trajectory by leveraging model's intrinsic capabilities. We introduce specific action tokens that enable the model to seamlessly backtrack and edit its own history within a single forward pass. By internalizing the revision loop, our framework Stream of Revision allows the model to activate its latent capabilities just-in-time without external dependencies. Empirical results on secure code generation show that Stream of Revision significantly reduces vulnerabilities with minimal inference overhead.

## 1. Introduction

Standard LLM code generation is typically cast as monotonic autoregressive decoding, where each token is considered immutable once it is generated (Vaswani et al., 2017; Jiang et al., 2024; Jain et al., 2025; Tian and Zhang, 2025; Wang et al., 2025; Huang et al., 2025). However, such a linear formulation stands in fundamental contrast to the

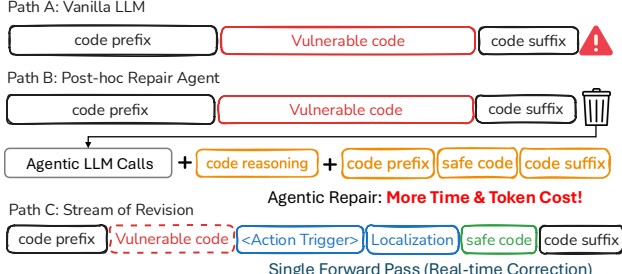

*Figure 1.* **Stream of Generation vs. Stream of Revision.** Conventional code generation treats generation as a linear stream of token appending, lacking the ability to revise earlier tokens. In contrast, our proposed Stream of Revision framework introduces action tokens that enable dynamic backtracking and in-place editing within a single pass.

cognitive process of programming. In practice, coding is rarely a single-shot trajectory; it is an interleaved process of generation and on-the-fly editing. Empirical studies of developer behavior support this view: Lubin et al. (Lubin and Chasins, 2021) observe that type construction for many programmers is not merely linear (or even branching) process, but fundamentally cyclical. Developers frequently perform just-in-time corrections, jumping back to edit and patch earlier lines immediately after spotting a potential flaw, rather than waiting for the entire function to be completed.

Unfortunately, the standard autoregressive architecture lacks native primitives to model this workflow. Specifically, conventional decoding provides no built-in mechanism for in-place rewriting and restricts decoding to monotonic token appending to an immutable prefix. As a result, revision intention is not naively expressible within the decoding process, forcing the model to treat each generated token as immutable ground truth implicitly. This limitation amplifies error propagation: early defects cannot be repaired when they are detected and tend to cascade into compounded downstream failures (Olausson et al., 2023).

To bridge this gap, existing approaches incorporate external components to enable LLMs to perform editing tasks (Madaan et al., 2023; Jiang et al., 2025; Yang et al., 2025). Some append a post hoc editing stage after the full trajectory is generated (Madaan et al., 2023; Liu et al., 2024; Yang et al., 2025), while others inject external analyzers into the decoding loop to trigger constrained regeneration (Wei

---

[1]School of Computing and Information Systems, Singapore Management University, Singapore [2]School of Cyber Science and Engineering, Huazhong University of Science and Technology, China [3]Monash University, Melbourne, Australia [4]National University of Singapore, Singapore. Correspondence to: Ming Wen <mwenaa@hust.edu.cn>.

*Proceedings of the $43^{rd}$ International Conference on Machine Learning*, Seoul, South Korea. PMLR 306, 2026. Copyright 2026 by the author(s).

et al., 2023; Jiang et al., 2025). While these methods provide evidence that LLMs possess *latent* self-repair abilities, they either incur substantial latency and token overhead or restrict intervention to syntactic constraints or static checks.

If repair is already within the model's capacity, why do current systems still externalize it to extra rounds or tools, rather than enabling revision *within a single decoding pass*, as developers do in practice? We address this by reformulating code generation as a decoding problem with *native revision actions*. We make revision an explicit part of the autoregressive output space, allowing the model to interleave generation with on-the-fly edits in a single pass. Specifically, we augment the model's vocabulary with a special revision-trigger token, expanding the output space into a hybrid domain of code generation and cursor manipulation. At each step, the model implicitly decides between continuing generation and initiating revision. Selecting the latter transitions the model into a constrained decoding state, where it emits a series of structured sequences of tokens specifying a revision operation to the already generated code (see the example in Fig. 1). By internalizing the revision loop, *Stream of Revision* equips LLMs with a *Virtual Cursor*, effectively allowing self-correction during a single forward pass without significant computational overhead. Finally, a deterministic renderer is applied to execute revision and produce an executable and user-friendly program.

To demonstrate the feasibility of this idea, we instantiate Stream of Revision on secure code generation, where just-in-time revision is especially high-leverage: a handful of tokens can prevent catastrophic vulnerabilities. More broadly, we consider revision as a structured decomposition of task-specific best practices into native decoding actions. For secure code generation, the best-practice revision workflow naturally includes three stages: vulnerability detection, localization, and repair, and we initialize this workflow directly as token primitives. Concretely, a revision trigger token serves as an intrinsic detection signal, followed by executable tokens that (i) localize the vulnerable span under constrained decoding and (ii) specify the patch to apply, enabling the model to pause and correct issues on the fly.

Crucially, we demonstrate that this mechanism activates the model's latent capabilities with data efficiency. We align the LLMs using revision trajectories extracted from real-world CVE records, and find that 1,000 positive examples suffice to elicit self-correction behavior. As a result, Stream of Revision substantially improves the security of generated code while preserving functionality, achieving matched or superior performance compared to recent secure-code baselines on the CyberSecEval 2 benchmark (Bhatt et al., 2024).

Our core contributions are summarized as follows: 1) We propose *Stream of Revision*, a decoding time reformulation that augments the output space with executable revision

tokens and compiles the generated trajectory into a final program via a deterministic renderer. 2) We operationalize secure coding revision as a three-stage episode with detection, localization under a strict substring constraint, and explicit patch synthesis. 3) We empirically show that self-correction is data efficient and that our method improves security on CyberSecEval while preserving utility.

## 2. Problem Formulation

### 2.1. Task Definition: Secure Code Generation

Let $\mathcal{X}$ denote the space of natural language specifications and $\mathcal{Y}$ the space of valid source code programs. The objective of standard code generation is to approximate a conditional distribution $p(Y|X)$ that maximizes the likelihood of functionally correct programs. In the context of *secure* code generation, we introduce a strict security constraint represented by an oracle $\mathcal{S} : \mathcal{Y} \rightarrow \{0, 1\}$. A generated program $Y$ is considered secure & correct if and only if:

$$\text{Secure \& Correct}(Y) \iff \text{Pass Test}(Y, X) \wedge (\mathcal{S}(Y) = 1)$$

where $\mathcal{S}(Y) = 1$ signifies that $Y$ is free from specific vulnerability patterns (e.g., CWEs).

### 2.2. The Autoregressive Bottleneck

Standard LLMs factorize generation autoregressively: $p(Y|X) = \prod_t p(y_t|y_{<t}, X)$. This formulation enforces a strict Identity Mapping ($Y \equiv T$) between the temporal generation trajectory $T = (t_1, \ldots, t_L)$ and the final spatial program $Y = (y_1, \ldots, y_L)$. This isomorphism imposes an *append-only* constraint: the action at step $t$ is irreversibly equal to the output at position $i = t$, which leads to fundamental limitations.

**The Immutability Constraint.** One fundamental limitation stems from the append-only nature of standard decoding: once a token is produced, it becomes part of an immutable prefix that conditions all future probabilities. Since real-world corpora inherently contain diverse vulnerability patterns, pre-training inevitably assigns non-zero probability mass to insecure spans. Consequently, regardless of the decoding policy (e.g., greedy or sampling), the model risks sampling a vulnerability. Crucially, lacking a native revision mechanism, such a locally insecure span, once emitted, instantly solidifies into a hard constraint that the model cannot retract within a single decoding process.

**The Commitment Trap.** Prefix immutability creates a phenomenon we term the *Commitment Trap*. Once an insecure span is generated, the autoregressive objective forces the model to treat this error as ground truth. Even if the model recognizes the vulnerability, it lacks the mechanism

to retract it. Instead, it is compelled to double down on the mistake, generating subsequent code that aligns with the error to maintain logical consistency. Consequently, a single local error cascades into a systemic failure (Olausson et al., 2023), simply because the model is trapped by its own immutable history.

## 3. Methodology

To address the immutability constraint in standard decoding, we represent the generated token sequence as an operation stream $T$ and compile it into a user-facing program $Y$ with a deterministic renderer $R$, i.e., $Y = R(T)$. Crucially, generation remains strictly monotonic: the model still emits a single autoregressive token stream, while non-monotonic edits are expressed as executable revision primitives interpreted by the renderer.

In this framework, revision is realized by embedding a small set of executable primitives directly into the token stream. Specifically, the operation stream $T$ interleaves normal code tokens with special edit tokens that encode in-place repair actions at the representation level. The Virtual Cursor provides a unified abstraction for these primitives, while the renderer deterministically executes them to produce the final program. This section details the representational primitives of this framework, the deterministic rendering mechanism, and the training and inference procedures.

### 3.1. Representational Primitives: The Virtual Cursor

We extend the standard vocabulary $\mathcal{V}_{\text{code}}$ with a set of operational tokens $\mathcal{V}_{\text{edit}}$, forming an augmented vocabulary $\mathcal{V} = \mathcal{V}_{\text{code}} \cup \mathcal{V}_{\text{edit}}$. These operational tokens act as executable control instructions for a *Virtual Cursor*, allowing the model to represent revision within a single autoregressive stream. To keep the mechanism lightweight and compatible with standard causal attention, we instantiate revision for secure code generation as three security-oriented operations: vulnerability *detection*, *localization*, and *patching* in the form of token-level primitives in $\mathcal{V}_{\text{edit}}$:

**I. Vulnerability Detection as Action Trigger ($\tau_{\text{trig}}$).** We trigger the decision to revise via a special token $\tau_{\text{trig}}$. At each decoding step $t$, the model computes $p(y_t \mid y_{<t})$ over the augmented vocabulary $\mathcal{V}$. Predicting $\tau_{\text{trig}}$ transitions decoding into a *Revision Episode*, illustrating that the current code prefix likely contains an insecure pattern that warrants auditing and correction. Conversely, predicting any token in $\mathcal{V}_{\text{code}}$ continues standard code generation.

**II. Vulnerability Localization via Content-Addressable Scope.** Upon triggering a revision episode, the model specifies the vulnerable span in the code prefix for modification. We adopt a **content addressable** localization strategy: the

model repeats the vulnerable span $s$ enclosed by scope delimiters, i.e., `<scope>`$s$`</scope>`. To prevent hallucinated localization, we enforce a **Strict Substring Invariant** during scope generation. Concretely, when generating $s$ given the current prefix $y_{<t}$, we apply dynamic logit masking so that the partial scope $(s \oplus v)$ must remain a valid substring of $y_{<t}$. Formally, for any token $v$, we enforce $p(v \mid y_{<t}) = 0$ if $(s \oplus v) \not\sqsubseteq y_{<t}$. This guarantees that the finalized scope is always a segment of the already generated prefix. (See Appendix A for the formal definition, transition rules, and exploration on alternative localization ways.)

**III. Vulnerability Repair via Explicit Patch Synthesis.** Given the localized span $s$, the model synthesizes a corrective patch $s'$, enclosed within `<patch>` delimiters. This representation explicitly conditions patch generation on the vulnerable code, enabling targeted refinement from the insecure span $s$ to the secure span $s'$.

**Revision Episode.** We serialize a revision episode as $\mathcal{E} = \tau_{\text{trig}} \oplus \langle\texttt{scope}\rangle s \langle/\texttt{scope}\rangle \oplus \langle\texttt{patch}\rangle s' \langle/\texttt{patch}\rangle$.

**Revision Trajectory.** The overall output stream remains strictly autoregressive: $T = y_{<t} \oplus \mathcal{E} \oplus y_{\text{resume}}$. Although $T$ is a single linear sequence, $\mathcal{E}$ encodes a non-monotonic edit on the prefix by specifying a localized scope $s \sqsubseteq y_{<t}$ and its replacement $s'$.

### 3.2. The Deterministic Renderer

The raw trajectory $T$ interleaves code tokens and revision artifacts, and is therefore not directly executable by standard compilers. We introduce a deterministic renderer $\Phi$ that acts as a stream interpreter. Given a revision episode specifying a scope $s$ and a patch $s'$, the renderer applies the edit tokens by replacing the targeted occurrence of $s$ in the current buffer and then appending $y_{\text{resume}}$ to produce the final program. It maintains a user-facing program buffer $B$ and processes the token stream sequentially with the following properties:

- **Stream Transparency.** During standard generation ($\tau \in \mathcal{V}_{\text{code}}$), tokens are immediately appended to $B$, and thus the streaming interface behaves like standard autoregressive decoding.

- **Opaque Repairing.** Upon encountering $\tau_{\text{trig}}$, the interpreter enters a hidden state. The subsequent localization $s$ and patch $s'$ are buffered internally and suppressed from the user-facing output, allowing the model to revise without exposing to end-users.

- **Atomic Commitment.** The revision is applied transactionally only upon observing the closure token `</patch>` only. The interpreter resolves the target by selecting the right-most occurrence of $s$ in the buffer $B$ as a deterministic tie-breaking rule, and performs an in-place

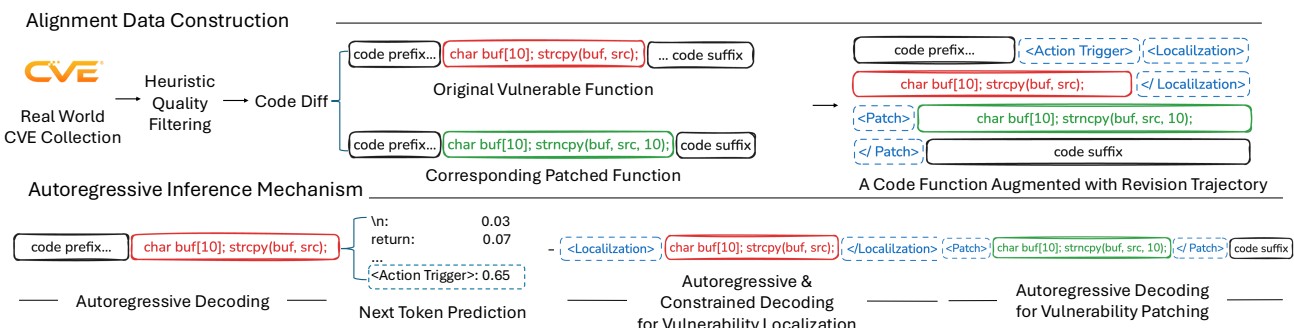

*Figure 2.* Overview of Stream of Revision for alignment data construction and single pass inference. Top: from real world CVE pairs, we filter, extract code diffs, and linearize the change into a revision trajectory with an revision trigger, a localized vulnerable span, and a patch span. Bottom: during autoregressive decoding, the model can emit a trigger token to start a revision episode, localize a vulnerable span under constrained decoding, and then generate a patch span. A deterministic renderer applies the patch atomically to the user facing buffer. Red highlights vulnerable code, green highlights patched code.

splice: $B \leftarrow B[: j^* - |s|] \oplus s' \oplus B[j^* :]$, where $j^*$ denotes the end index of the matched span $s$ in $B$. This operation updates $B$ from the pre-revision state to the patched state.

### 3.3. Training Strategy and Optimization

#### 3.3.1. SEMANTIC EMBEDDING INITIALIZATION

A common bottleneck in augmented-vocabulary training is the cold start problem of newly added tokens: randomly initialized embeddings often lead to unstable optimization and slow convergence, especially when the tokens correspond to structured control behaviors. To mitigate cold-start effects, we initialize each special token's embedding using the weighted average of its natural-language semantic description. This anchors the operational tokens in a meaningful region of the embedding space from the onset of training, aligning them with their intended behaviors. The full initialization procedure is provided in Appendix D.

#### 3.3.2. OPTIMIZATION OBJECTIVES

Supervised Fine-Tuning (SFT). We first adapt the model to the syntax of revision episodes. Given a dataset of augmented trajectories $\mathcal{D}_{\text{SFT}} = \{(x, T)\}$, where $T$ includes interleaved code and revision actions, we optimize the standard autoregressive negative log-likelihood: $\mathcal{L}_{\text{SFT}} = -\mathbb{E}_{(x,T)\sim\mathcal{D}_{\text{SFT}}} \sum_t \log P_\theta(v_t \mid x, v_{<t})$. This stage establishes the fundamental mapping from detection to repair, ensuring the model learns to construct syntactically valid scopes and patches. We also explore other alignment objectives like DPO and SimPO and detailed in Appendix D.

### 3.4. Efficiency Gain of Stream of Revision.

We analyze the computational efficiency of repair paradigms with respect to the input context length $L$. Let $\mathcal{C}(\cdot)$ denote the computational cost in terms of token processing.

**Scaling efficiency on Local Repairs.** For a localized vulnerability where the repair length is independent of the context length $L$ (i.e., $|y_{patch}| \ll L$), the computational overhead of retroactive repair agents scales linearly with $L$ ($\mathcal{O}(L)$), whereas the overhead of Stream of Revision is constant ($\mathcal{O}(|y_{patch}|)$), which approximates ($\mathcal{O}(1)$). Specifically, the standard post-hoc requires re-encoding the full context $L$ to re-generate the patched code, incurring a cost proportional to $L$. In contrast, Stream of Revision performs intervention during the initial pass. For a local patch of constant size, the marginal cost is proportional to $|y_{patch}|$, which is independent of $L$. (See Appendix A for formal proof).

## 4. Alignment Dataset Crafting

To equip a standard LLM with revisable decoding behavior, we align it to map a natural language specification $X$ to an augmented trajectory $\mathcal{T}$ that interleaves normal code tokens with revision episodes. Each revision episode operationalizes a minimal security editing lifecycle with three components: an action trigger, a localized scope span $s$, and a patch span $s'$. Full details is available at Appendix F.

To evaluate whether the learned behavior reflects security reasoning rather than language specific memorization, we align Stream of Revision using only C and C++ vulnerability pairs and evaluate the resulting revision capability on a multilingual suite of unseen languages (Cassano et al., 2023a; Niu et al., 2023).

### 4.1. Data Sourcing from Real World CVE Pairs

We construct alignment trajectories from real world CVE records by extracting paired vulnerable and patched function code from 9 public vulnerability benchmarks (Bhandari et al., 2021; Wang et al., 2024; Ding et al., 2025; CVE Project, 2025; NIST, 2025; GitHub, Inc., 2025). For each CVE, we obtain a ground truth code diff between a vulnera-

*Table 1.* Summary statistics of the experimental datasets.

|  | # Nums | # CWE | Avg. Tokens / Func. |
|---|---|---|---|
| Strict Samples | 1,022 | 105 | 1,017.62 |
| Relaxed Samples | 3,439 | 115 | 897.39 |

ble revision and its corresponding remediation patch.

**Function Level Extraction.** We decompose each commit into modified functions and construct a training example per affected function. For a modified function $f$, we derive a revision episode $\mathcal{E}$ by concatenating 1) base context: the vulnerable function body used as the initial code prefix for generation; 2) localization span $s$: the vulnerable code span modified by the remediation diff; 3) patch span $s'$: the replacement span from the patched version. This construction grounds every trigger $\tau_{\text{trig}}$ in a verified remediation signal and provides direct supervision for localizing and patching.

**Security Neutral Specifications.** To pair each trajectory with a natural language specification, we use a teacher model (Gmeini-3-pro) to generate $X$ that describes only the shared functional intent of the vulnerable and patched versions. The teacher is instructed to remain neutral about security and avoid mentioning vulnerability-related cues, so that the student must rely on its auditing and revision behavior rather than prompt shortcuts. The dataset statistic is in Table 1.

### 4.2. Data Quality Control

A key challenge in real-world CVE commits is **commit entanglement**, where security fixes co-occur with unrelated refactoring or feature updates. Such noise reduces the purity of revision supervision and empirically leads to excessive triggering at inference time. We therefore construct two tiers of alignment data based on commit granularity: 1) strict set: extracting data from CVE commits that modify exactly one function with exactly one code hunk; 2) relaxed set: extracting from commits that touch multiple functions and multiple hunks, which increases coverage but might introduce noise. We use the strict set as the primary supervision source and the relaxed set to investigate the quality-scale tradeoff.

### 4.3. General Instruction Replay

To prevent catastrophic forgetting and reduce false triggering on benign code, we mix the revision trajectories with a high-quality general code instruction corpus (Luo et al., 2023). We keep a fixed mixing ratio $\lambda$ between revision data and general instructions and filter the general corpus to prioritize C and C++ code blocks. This exposes the model to both vulnerable and benign C and C++ contexts, helping decouple language syntax from the revision signal.

## 5. Experiment Setting

### 5.1. Evaluation Dataset

We evaluate Stream of Revision on two distinct categories of benchmarks to assess both its proactive security capabilities and its preservation of general coding utility. **Security Benchmarks.** To measure the model's ability to defend against vulnerability-inducing prompts, we employ CyberSecEval 2 (Bhatt et al., 2024) (CSE2) by Meta as our primary testbed. **Utility Benchmarks.** To ensure the alignment process does not degrade general programming proficiency, we evaluate functional correctness on HumanEval on C/C++ (Cassano et al., 2023b).

### 5.2. Baselines

There are no direct predecessors of Stream of Revision. To rigorously evaluate our performance, we compare against representatives from three adjacent paradigms: (1) vanilla LLMs, (2) coding agents that performing vulnerability detection and repair during inference stage (full details of both agents are available at Appendix G), and (3) LLMs with preference alignment on security, SafeCoder (He et al., 2024) and ProSEC (Xu et al., 2024). Notably, Stream of Revision is orthogonal to the alignment methods mentioned above. While methods like ProSec optimize the base distribution to reduce the likelihood of vulnerabilities, our approach provides a runtime safety guard to intercept them when they do occur. Thus, Stream of Revision can be integrated on top of security-aligned models to further enhance robustness.

### 5.3. Evaluation Metric

Following the ProSec protocol (Xu et al., 2024), we evaluate general coding utility with Pass@1 & Pass@10 metric. To assess secure code generation capabilities, we adopt the evaluation setup from CyberSecEval 2 (Bhatt et al., 2024). For each test instruction in CSE2, we generate multiple samples and calculate the ratio of secure code among all generated samples using static checkers provided by CSE2. Furthermore, to quantify the impact of our renderer and Stream of Revision's affection to code syntax, we calculate the AST parser success rate of the final generated code. This serves as a proxy for the model's ability to preserve syntactic correctness while performing internal revisions.

### 5.4. Implementation Details

Full implmentation details are provided in the Appendix E.

## 6. Experiment Result

We evaluate the effectiveness of Stream of Revision in mitigating vulnerabilities. Table 2 compares the performance across different LLMs and programming languages.

*Table 2.* **Main Results.** Comparison of Stream of Revision against baselines across multiple backbones. **Security Effectiveness** reports the Security Pass Rate (SPR@1, ↑) on **CyberSecEval 2 (CSE2)**. We distinguish between the **Target Domain** (C and C++, seen during alignment of Stream of Revision) and **Generalization** (Python, Java, JavaScript, C#, untrained in Stream of Revision but under trained by Prosec). Stream of Revision achieves superior defense in C/C++ while demonstrating significant zero-shot transfer to OOD languages without compromising utility.

| Backbone | Method | Security Effectiveness (CSE2, SPR@1 ↑) | | | | | | | | Functionality (↑) | |
| | | Target Domain | | | Generalization (OOD) | | | | | HumanEval* | |
| | | C | C++ | Avg | C# | Java | JS | Python | All Avg | Pass@1 | Pass@10 |
|---|---|---|---|---|---|---|---|---|---|---|---|
| **Qwen2.5-7B** | Base Model | 64.36 | 76.03 | 70.20 | 72.48 | 61.59 | 59.84 | 75.18 | 68.24 | 50.80 | 81.98 |
| | SafeCoder | 73.76 | 82.64 | 78.20 | 70.64 | 68.92 | 58.61 | **85.46** | 73.33 | 53.71 | 83.22 |
| | Prosec | 73.26 | 79.75 | 76.51 | 66.06 | 65.24 | 63.52 | 82.27 | 71.68 | 44.16 | 77.64 |
| | **Stream of Revision** | **73.76** | **83.47** | **78.62** | **79.81** | **70.12** | **65.16** | 81.56 | **75.65** | 54.78 | 79.50 |
| **Qwen2.5-3B** | Base Model | 62.38 | 76.45 | 69.42 | **77.98** | 62.80 | 61.84 | 71.50 | 68.83 | 57.32 | 82.60 |
| | SafeCoder | 67.84 | **86.48** | 77.16 | 75.74 | 54.14 | 61.84 | 81.76 | 71.30 | 59.42 | 83.14 |
| | Prosec | 64.75 | 81.08 | 72.92 | 67.65 | 55.02 | **73.89** | 83.47 | 70.98 | 58.91 | 84.47 |
| | **Stream of Revision** | **75.25** | 84.30 | **79.78** | 73.85 | **69.51** | 68.44 | 83.69 | **75.84** | 59.62 | 82.61 |
| **CodeLlama-7B** | Base Model | 70.40 | 78.93 | 74.67 | 75.50 | **76.10** | **70.82** | 83.90 | 76.58 | 29.19 | 56.52 |
| | SafeCoder | 75.25 | 86.12 | 80.69 | 75.41 | 75.85 | 67.70 | **86.31** | 77.77 | 31.71 | 59.71 |
| | Prosec | 69.01 | 80.58 | 74.80 | 75.69 | 74.88 | 69.75 | 83.62 | 75.98 | 30.00 | 52.79 |
| | **Stream of Revision** | **77.23** | **86.37** | **81.80** | **77.52** | 72.68 | 69.39 | 85.11 | **78.05** | 30.90 | 62.73 |

*Table 3.* **Efficiency and Security Comparison on CSE2.** We compare the average token consumption per problem (Input and Output) and the Security Pass Rate (SPR) across C and C++ languages. Stream of Revision achieves comparable or superior security to the Post-hoc Agent while significantly reducing computational cost.

| Method | Avg. Token Usage (↓) | | Security Pass Rate (↑) | |
| | Input | Output | C | C++ |
|---|---|---|---|---|
| **Qwen2.5-7B (Vanilla)** | 113.10 | 205.37 | 64.36% | 76.03% |
| **Post-hoc Agent w/o localization** | 558.67 | 426.33 | 67.86 | 79.75 |
| **Post-hoc Agent with localization** | **742.53** | **480.99** | 70.87 | 82.23 |
| **Qwen2.5-7B (SFT on Patch only)** | 113.10 | 207.04 | 66.48% | 78.17% |
| **Stream of Revision (Ours)** | 113.10 | 219.90 | **73.76** | **83.47** |

**Superiority in Target Domain.** Our primary evaluation focuses on C and C++, the languages used for training the revision mode. Stream of Revision shows on-par or superior performance compared to all baselines across different backbones on C and C++. Specifically, Stream of Revision outperforms vanilla LLMs on C/C++ by 9.55%–14.92% across base models. This confirms that the internal revision mechanism effectively enhances the model's ability to generate secure code in the target domain. Notably, Stream of Revision consistently performs significantly (p<0.01) better than vanilla LLMs, indicating the effectiveness of our revision machinism to correct vulnerabilities during generation.

Furthermore, in the Top-10 CWE from the 2025 Top-25 most dangerous software weaknesses list (The MITRE Corporation, 2025), Stream of Revision improves the average SPR from 71.6% to 78.7% (+7.1%) and provides consistent gains across various vulnerability types; detailed results are provided in Appendix H.

**Zero-shot Cross-lingual Generalization.** We observe that, while Stream of Revision was trained exclusively on C/C++

data, it demonstrates remarkable zero-shot transfer capabilities to all out-of-distribution (OOD) languages. Notably, in C# and Python, our method remains competitive with or superior to baselines that likely included these languages in their alignment data. This suggests that Stream of Revision learns *language-agnostic* security patterns (e.g., identifying unsafe input handling or buffer risks) rather than simply overfitting to the syntax of the training languages.

This comparison strengthens our claim of generality: matching the performance of fine-tuned security-aligned models on OOD languages indicates that Stream of Revision is not merely overfitting to C/C++ syntax, but instead internalizes *language-agnostic* vulnerability patterns that transfer across programming languages. Specifically, averaged over all programming languages, Stream of Revision outperforms the best-performing baselines by 0.36%–6.37% across base models. We emphasize that our goal is not to outperform specialized baselines on every language, since those methods may have benefited from training supervision in the evaluated languages. Rather, our objective is to show that a lightweight in-decoding revision interface, learned from a narrow C/C++ supervision set, can generalize robustly beyond the training distribution.

**Utility Preservation.** We also identify that training with Stream of Revision does not compromise the model's general coding capabilities. Across all backbones, Stream of Revision maintains or slightly improves Pass@1 and Pass@10 on HumanEval compared to both the base models and other alignment baselines. Notably, we observe in HumanEval, where most coding tasks have minimal security threats, the time of revision trigger is also minimal. The tuned Qwen2.5-7B triggers a revision in 19.16% of CSE2 while only once

in HumanEval, which is a pure code refactoring.

**Qualitative Observation on Revision.** We qualitatively analyze Stream of Revision's revision behavior by manually inspecting a subset of 40 revised samples on CSE2. We observe two recurring patterns. First, Stream of Revision often identifies and tries to patch vulnerabilities or buggy patterns during generation, including buffer overflows, unsafe input handling, and memory safety issues (see Appendix I). Second, one-third of revisions are not directly related to security patching. These revisions are mainly localized edits for code refactoring. We do not observe obvious regressions in security or functionality in sampled revisions for code refactoring, however, those edits still introduce additional yet light computational overhead and latency.

## 7. Analysis

### 7.1. Efficiency Analysis

**Computation Efficiency.** We evaluate computational efficiency against vanilla LLM and two constructed post-hoc agent baselines following Agentless (Xia et al., 2024), which performs sequential code generation, vulnerability detection, localization, and repair (Appendix C).

Table 3 demonstrates that Stream of Revision achieves security performance on par with the post-hoc agent while drastically reducing computational costs. By operating in a single pass, Stream of Revision eliminates the redundant context processing of agentic workflows, resulting in a $6.5\times$ reduction in input tokens (113.10 vs. 742.53). On the other hand, compared to the vanilla LLM, Stream of Revision incurs only a $1.06\times$ increase in output tokens. This validates that high-security generation does not require the prohibitive $\mathcal{O}(L)$ latency of iterative repair.

**Training Efficiency and Data Quality.** We investigate the impact of training data scale on model performance. We compare Qwen-2.5-7B, fine-tuned on a high-quality subset of 923 positive samples, against a variant trained on an expanded dataset of over 8,000 positive samples (see Sec 4.2). To isolate the effectiveness of the revision mechanism, we introduce an ablation variant trained on the same dataset as Stream of Revision, but excluding the intermediate revision sequences. This model is trained to directly generate the final patched functions. We measure cumulative token usage (input and output) on CSE2 using Qwen2.5-7B.

As illustrated in Figure 3, both models achieve statistically indistinguishable SPR across C and C++. However, the model trained on the larger corpus exhibits a significant increase in inference token consumption due to more frequent revision triggering by code refactoring, likely due to data noise in training. This finding underscores that for learning the revision mechanism, data quality and precision are

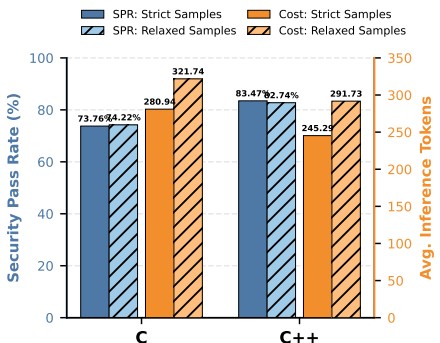

*Figure 3.* **Impact of Training Data Scale.** **Blue bars (Left Axis)** denote Security Pass Rate (SPR), **Orange bars (Right Axis)** denote Avg. Inference Tokens. Comparing the hatched bars to solid bars shows that adding more data yields **negligible security gains** but incurs **higher inference costs** due to more revisions.

paramount over raw quantity, and our method is sample-efficient. Meanwhile, the ablation variant without revision sequences only yields marginal security improvements over the vanilla model and falls short of Stream of Revision by a wide margin (-7.28% C, -5.30% C++), underscoring the effectiveness of the revision mechanism.

### 7.2. Syntactic Validity

We assess the syntactic stability of Stream of Revision's revisions by examining compilation outcomes (using AST parsing as proxy) before and after revision episodes for tuned Qwen2.5-7B on CSE2. Table 4 presents a confusion matrix of compilation status. There are 322 data points out of 1681 total (19.16%) where revisions were triggered. 98.45% of revision operations are non-destructive, preserving the compilation status (green and gray cells). Only 1.55% of revisions lead to regression (red cell) or fixing (purple cell), indicating that Stream of Revision maintains high syntactic integrity during self-correction.

We also evaluate the effectiveness of Stream of Revision's constrained decoding by replacing substring-constrained vulnerability localization with an unconstrained variant. While constrained localization guarantees that the localized span is always a substring of the generated prefix, unconstrained localization hallucinates non-existent spans in 10.14% of cases (36/355).

*Table 4.* **Syntactic Stability.** Confusion matrix of compilation status for revised code. Cells show count and category. **Green/Gray** indicates stability; **Red** indicates regression. The high diagonal values indicate that our method is non-destructive.

|  |  | Post-Revision | |
|---|---|---|---|
|  |  | **Pass** | **Fail** |
| **Pre-Revision** | **Pass** | **258** (Stable) | **4** (Regressed) |
|  | **Fail** | 1 (Fixed) | 59 (Stable) |

Table 5. **Complementarity with Alignment Approaches.** We apply Stream of Revision on top of ProSec. The combination yields performance gains, demonstrating that our inference-time revision is orthogonal to weight-level alignment methods.

| Method | Security Pass Rate (C) | Security Pass Rate (C++) |
|---|---|---|
| **ProSec** (Baseline) | 73.26% | 79.75% |
| **ProSec + SoR (Ours)** | **74.42%** | **83.90%** |

### 7.3. Stream of Revision together with aligned models

Current SOTA methods like ProSec employ alignment techniques like DPO to shift the model's global distribution towards security. A critical question is whether Stream of Revision is redundant to these efforts or complementary.

As shown in Table 5, applying Stream of Revision on top of the ProSec checkpoint further boosts security performance (C: +1.16%, C++: +4.15%). This suggests that weight-level alignment and inference-time revision operate on orthogonal axes. By combining both, we achieve a synergistic effect that surpasses what either method can achieve in isolation. Furthermore, we evaluate the performance of aligning Stream of Revision with DPO and SimPO and discuss in Appendix D.

### 7.4. Effectiveness of Logit Calibration

We investigate the controllability of Stream of Revision by manipulating the generation probability of the action trigger token (i.e., the revision initiator). By introducing a scalar bias to the logits of the trigger token, we can adjust the model's sensitivity to potential vulnerabilities.

We observe a distinct trade-off between performance and computational cost. Increasing the trigger probability from 1 to 10 leads to a significant increase in computational overhead due to frequent revisions (e.g., $1.54\times$ tokens on Qwen2.5-7B). In terms of performance, this aggressive intervention strategy yields a minimal improvement in security correctness (e.g., +1.41% on Qwen2.5-7B). A larger bias would lead to LLM output collapse. Considering the computational budget, the marginal performance gains do not justify the exponential growth in token consumption. Therefore, we adopt a balanced calibration in our main experiments to maximize security coverage while maintaining inference efficiency.

### 8. Related Work

**Aligning LLMs for Code Generation.** (Guan et al., 2024) distill reasoning traces for training secure code generation. (He et al., 2024) employs a security-centric fine-tuning with instruction tuning to enhance code safety. (Xu et al., 2024) utilizes Direct Preference Optimization (DPO) to align LLMs towards secure coding practices and an automatic data pipeline to synthesize security datasets. (Hasan et al., 2025) introduce an localized preference optimization method to fine-tune LLMs for secure code generation on vulnerability spans. Our approach is orthogonal to these weight-level alignment methods, focusing on 1) enhancing the inference-time revision capabilities of LLMs and 2) mainly leveraging the intrinsic capacity of LLMs.

**In-decoding Intervention of Code Generation.** Rocode (Jiang et al., 2025) integrates static analyzers into the decoding loop, triggering rollback and constrained regeneration upon detecting syntactic violations. (Wei et al., 2023) introduces an automatic code completion tool in constrained decoding for effective program repair. JumpCoder (Chen et al., 2024) relies on a separate infilling model and external judges to retrospectively insert missing code lines. Differently, our method internalizes revision as a first-class generation action via learned edit tokens, enabling a single model to perform online revision for secure code generation. (Zheng et al., 2023) allows for postponing the generation of specific code until a definitive suffix is established, boosting code generation ability of LLM. GCD (Park et al., 2025) applies the grammar of programming languages to guide code generation with constrained decoding. While those approaches perform in-decoding intervention, they rely on external rule-based tools for syntax-level error detection and localization, which limits their ability to handle complex semantic vulnerabilities. In contrast, our Stream of Revision framework empowers LLMs with intrinsic semantic capabilities, enabling just-in-time vulnerability detection, localization, and repair without external dependencies.

There are some edit-based, non-autoregressive, and diffusion-based models that support iterative editing or denoising (Singh et al., 2023; Xie et al., 2025). While this enables global refinement, diffusion-style paradigms typically maintain a full-sequence representation and perform multiple global denoising updates, which is not natively support left-to-right token streaming and often incurs more overhead. In contrast, Stream of Revision remains fully autoregressive: it preserves a single left-to-right stream and encodes revision as discrete actions that are executed by a deterministic renderer.

### 9. Conclusion and Future Work

In this work, we challenged the monotonic rigidity of standard autoregressive decoding. We introduced Stream of Revision, a framework that internalizes the "draft-and-revise" process directly into the generation stream. By redefining code generation as a dynamic trajectory $Y = \Phi(T)$ rather than a static sequence, we enabled models to perform just-in-time revision, correcting latent vulnerabilities on-the-fly without exposing the revision mechanics to the end-user. We empirically demonstrate that Stream of Revision bridges

the gap between the speed of single-pass generation and the precision of agentic repair. It achieves on-par or superior security performance with SOTA approaches while maintaining constant inference overhead.

## Acknowledgments

This research / project is supported by the National Natural Science Foundation of China (No.62372193 and No.U2436207), and the National Research Foundation, under its Investigatorship Grant (NRF-NRFI08-2022-0002). Any opinions, findings and conclusions or recommendations expressed in this material are those of the author(s) and do not reflect the views of National Research Foundation, Singapore.

## Impact Statement

This paper presents work whose goal is to advance the field of Machine Learning, with a particular focus on improving the reliability and safety of code generation models. By enabling just-in-time revision within a single autoregressive decoding stream, our method aims to reduce the incidence of insecure or erroneous code produced by large language models, thereby lowering downstream risks in software development.

The primary societal impact of this work is positive: more trustworthy code generation systems can help developers avoid common security pitfalls and improve software quality, which is especially important as LLMs are increasingly used in real-world programming workflows. At the same time, the proposed mechanism introduces a more powerful form of in-decoding control, which, if misused or improperly calibrated, could lead to unintended modifications or opaque behavior during generation. We therefore emphasize that such systems should be deployed with appropriate safeguards, logging, and evaluation, particularly in safety-critical settings.

Overall, we believe that the ethical and societal implications of this work are aligned with the well-established goals of improving the robustness and safety of machine learning systems, and we do not foresee significant negative consequences beyond those already common to the deployment of large language models.

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

# A. Formalization of Trie-Based Constrained Decoding

In this section, we provide the mathematical formulation of the constrained decoding mechanism used in the Localization phase.

**Setup.** Let $y_{<t} = (y_1, \ldots, y_{t-1})$ denote the code prefix generated prior to the revision trigger. During localization, the model emits a span $s = (s_1, \ldots, s_k)$ that is required to be a contiguous substring of $y_{<t}$. We write $a \sqsubseteq b$ to denote that $a$ is a contiguous subsequence of $b$.

**Match-Set Maintenance.** We maintain the set of feasible match end positions for the current partial span $s$:

$$I(s) \triangleq \{ j \mid |s| \leq j < t, \ y_{j-|s|:j} = s \}. \tag{1}$$

By definition, $I(s) \neq \emptyset$ if and only if $s \sqsubseteq y_{<t}$. This is equivalent to traversing the substring trie of $y_{<t}$, where $I(s)$ represents the active match states.

**Dynamic Vocabulary Masking.** At each step, we mask the model's output distribution to a dynamic valid set $\mathcal{V}_{\text{valid}}(s, y_{<t})$.

**Initialization** ($|s| = 0$): Any token present in the prefix can start a match:

$$\mathcal{V}_{\text{valid}}(\emptyset, y_{<t}) = \{ y_i \mid 1 \leq i < t \}. \tag{2}$$

**Continuation** ($|s| > 0$): A token $v$ is a valid continuation if it extends at least one active match in $I(s)$:

$$\mathcal{V}_{\text{cont}}(s, y_{<t}) \triangleq \{ y_j \mid j \in I(s), \ j < t \}. \tag{3}$$

We additionally allow a scope-closure token $\tau_{\text{end}}$ to terminate the localization scope:

$$\mathcal{V}_{\text{valid}}(s, y_{<t}) = \mathcal{V}_{\text{cont}}(s, y_{<t}) \cup \{\tau_{\text{end}}\}, \quad \text{for } |s| > 0. \tag{4}$$

**State Update Transition.** When the model emits a token $v \in \mathcal{V}_{\text{cont}}(s, y_{<t})$, the match set updates as:

$$I(s \oplus v) = \{ j + 1 \mid j \in I(s), \ j + 1 < t, \ y_j = v \}. \tag{5}$$

The localization phase terminates when $\tau_{\text{end}}$ is emitted.

**Disambiguation.** Upon termination with a finalized span $s$, if $|I(s)| > 1$, the executor resolves ambiguity by selecting the right-most occurrence:

$$j^* = \max(I(s)). \tag{6}$$

The localized target window is then the half-open interval $[j^* - |s|, j^*)$ in the prefix, i.e., $y_{j^*-|s|:j^*} = s$. We choose the right-most occurrence over the left-most, since the vulnerability detection trigger is intuitively associated with the most recent and thus right-most code span.

**Alternative Localization** We explored an alternative strategy that prompts the LLM to explicitly output the line numbers corresponding to the vulnerable span (e.g., start_line and end_line). However, we empirically found this approach to be highly unstable. A frequent failure mode is hallucination, where the model predicts coordinates that do not correspond to the actual code structure or do not match the corresponding fix content. In severe cases, the predicted line numbers even exceed the total length of the generated function. We argue that this instability stems from a misalignment with the LLM's internal representation: models are trained on token sequences, not line-indexed buffers. Consequently, counting lines deviates from the model's pre-training distribution, rendering this prediction unreliable without fine-tuning on a larger dataset.

**Localization Range** Our method may be less effective for localizing vulnerabilities that require asynchronous edits across multiple hunks. Nonetheless, the setting remains practically important: in existing datasets from CVE records, more than half of vulnerability fixes can be resolved with a single hunk modification in each function. We focus on addressing these common failure modes rather than attempting to handle every vulnerability type.

# B. Efficiency Benefits of Stream of Revision.

Assumption: Localized Vulnerability. We define a "Local Vulnerability" as a case where the length of the required secure patch $N_s$ is negligible compared to the context length $L$ (i.e., $N_s \ll L$) and does not grow as $L$ increases. This covers a vast number of code completion security issues (e.g., API misuse, insecure defaults, missing checks).

While global refactoring is outside the scope of real-time completion, many security flaws introduced during coding (e.g., strcpy, SQL injection) are inherently local. For these high-frequency scenarios, our $\mathcal{O}(1)$ overhead provides a decisive advantage over $\mathcal{O}(L)$ agentic methods.

## B.1. Notations and Assumptions

Let $\mathcal{C}(\cdot)$ denote the computational cost (e.g., FLOPs or token processing time) of a Transformer-based decoder. We define the following variables:

- $x$: The input context (prompt + code prefix) with length $L = |x|$.

- $y_{vul}$: The vulnerable code segment generated by a standard model, with length $N_v$.

- $y_{sec}$: The secure patch required to fix the vulnerability, with length $N_s$.

**Assumption 1 (Local Vulnerability).** We focus on localized security issues (e.g., API misuse, boundary checks) where the length of the required patch is negligible compared to the context and does not grow with it. Formally, we assume $N_s, N_v \ll L$ and $N_s \approx \mathcal{O}(1)$.

## B.2. Complexity of Post-hoc Repair Agents

The standard agentic repair process operates in a "Generate-then-Repair" cycle, consisting of two distinct phases:

**Phase 1 (Generation):** The model processes context $x$ and generates the vulnerable candidate $y_{vul}$.

$$\mathcal{C}_{\text{gen}} = L + N_v \tag{7}$$

**Phase 2 (Repair):** To generate the fix, the agent must re-encode the original context $x$ along with the error $y_{vul}$ (and typically a system instruction) to condition the repair generation.

$$\mathcal{C}_{\text{repair}} = (L + N_v) + N_s \tag{8}$$

The total cost is the sum of both phases:

$$\mathcal{C}_{\text{agent}} = \mathcal{C}_{\text{gen}} + \mathcal{C}_{\text{repair}} = 2L + 2N_v + N_s \tag{9}$$

**Overhead:** The computational overhead relative to an ideal single-pass generation ($L + N_s$) is dominated by the redundant re-encoding of $L$:

$$\Delta_{\text{agent}} = \mathcal{C}_{\text{agent}} - (L + N_s) \approx L + 2N_v \implies \mathcal{O}(L) \tag{10}$$

## B.3. Complexity of Stream of Revision

Our approach utilizes a single forward pass with latent intervention:

**Phase 1 (Prefix Processing):** The model processes context $x$ once.

$$\mathcal{C}_{\text{pre}} = L \tag{11}$$

**Phase 2 (Intervention):** Upon detecting the latent vulnerability, the model triggers the `<revision>` token (cost $\approx 1$) and immediately switches to generating the secure patch $y_{sec}$.

$$\mathcal{C}_{\text{fix}} = 1 + N_s \tag{12}$$

The total cost is:

$$\mathcal{C}_{\text{ours}} = L + 1 + N_s \tag{13}$$

**Overhead:** The overhead relative to an ideal generation is:

$$\Delta_{\text{ours}} = \mathcal{C}_{\text{ours}} - (L + N_s) = 1 \implies \mathcal{O}(1) \tag{14}$$

### B.4. Conclusion

Comparing the two paradigms as the context length scales ($L \to \infty$):

$$\lim_{L \to \infty} \frac{\Delta_{\text{agent}}}{\Delta_{\text{ours}}} = \lim_{L \to \infty} \frac{\mathcal{O}(L)}{\mathcal{O}(1)} = \infty \tag{15}$$

This derivation proves that under the assumption of local vulnerabilities, the overhead of post-hoc agents grows linearly with context length, whereas **Stream of Revision** maintains a constant overhead, ensuring scalability for long-context software engineering tasks.

## C. Detailed Experimental Setup for Efficiency Analysis

To rigorously evaluate the efficiency of Stream of Revision, we construct two distinct post-hoc repair baselines representing common agentic workflows. These baselines operate sequentially: generating code first, then performing security analysis, and finally executing repairs.

### C.1. Baseline Configurations

We compare Stream of Revision against the following two agentic paradigms. Both agents are constructed following the design principles of representative agent frameworks, Agentless (Xia et al., 2024). The details of agent construction and the selection justification are available at Appendix .

**Baseline I: Global Repair Agent (Classification + Repair).**  This baseline represents a coarse-grained repair approach, often seen in standard re-prompting iterative refinement.

1. **Generation:** The model generates the full code solution $y_0$ based on the problem description $x$.

2. **Classification:** An external critic (or the model itself) evaluates $y_0$ to determine if it contains vulnerabilities (Binary Classification: Secure/Vulnerable).

3. **Global Repair:** If marked vulnerable, the model is provided with the original code and a prompt to "fix the security issue," resulting in a complete regeneration of the function $y_{fix}$.

**Cost Implication:** This approach incurs high output token costs as it often rewrites the entire function, doubling the generation cost in the worst case.

**Baseline II: Localized Repair Agent (Classification + Localization + Repair).**

1. **Generation:** The model generates the full code solution $y_0$.

2. **Localization:** The critic identifies specific vulnerable lines (e.g., `line 12:  gets() is unsafe`).

3. **Localized Repair:** The model is prompted with the code and the specific error location. It generates a patch or a specific replacement for the identified lines only, rather than the whole function.

**Cost Implication:** While this minimizes *output* tokens (generating only the patch), it drastically increases *input* tokens. The model must re-read the full context and original code multiple times (once for localization, once for repair), leading to the $\mathcal{O}(L)$ complexity scaling discussed in Section 7.1.

## C.2. Token Usage Calculation

We calculate the Cumulative Token Usage ($\mathcal{C}_{total}$) for each problem as the sum of all input and output tokens across all turns until a secure solution is found or the maximum turn limit ($T = 3$) is reached.

For a multi-turn agentic baseline, the cost is defined as:

$$\mathcal{C}_{total} = \underbrace{|x| + |y_0|}_{\text{Initial Gen.}} + \sum_{t=1}^{T} \left( \underbrace{|x| + |y_{t-1}| + |p_{critic}|}_{\text{Input (Re-reading)}} + \underbrace{|y_{fix}|}_{\text{Output (Repair)}} \right) \tag{16}$$

where $p_{critic}$ represents the prompt overhead for the critic/repair instructions.

In contrast, our Stream of Revision operates in a single pass. Its cost is strictly:

$$\mathcal{C}_{Stream of Revision} = |x| + |y_{final}| + \underbrace{N_{rev} \times 1}_{\text{Intervention Tokens}} \tag{17}$$

where $N_{rev}$ is the number of small revision steps (typically $< 1\%$ of total tokens). This mathematical formulation highlights why Stream of Revision achieves massive token savings: it eliminates the $\sum(|x| + |y_{t-1}|)$ re-reading cost term entirely.

# D. Training Design

## D.1. Semantic Initialization

We introduce five operational tokens to support Stream of Revision: `<|backtracking|>`, `<|OLD|>`, `<|/OLD|>`, `<|NEW|>`, and `<|/NEW|>`. Each token is associated with a concise natural-language description:

- `<|backtracking|>` (the action trigger): "Triggers a backtracking operation to return to a previous context point."

- `<|OLD|>` (localization delimiter): "Localizes and quotes the specific vulnerable code segment from the context."

- `<|/OLD|>` (end localization delimiter): "Ends the localization of the vulnerable segment."

- `<|NEW|>` (patch delimiter): "Specifies the repaired code solution to substitute the localized vulnerability."

- `<|/NEW|>` (end patch delimiter): "Ends the repaired code definition."

Let $E(\cdot)$ denote the base model's embedding function. For each special token $\tau$, we construct its initial embedding as a weighted average over the embeddings of the words in its semantic description:

$$\mathbf{e}_\tau = \frac{1}{Z} \sum_{w \in \mathcal{D}_\tau} \alpha_w \, E(w), \quad Z = \sum_{w \in \mathcal{D}_\tau} \alpha_w,$$

where $\mathcal{D}_\tau$ is the set of tokens obtained by tokenizing the description of $\tau$, and $\alpha_w$ is a weighting factor (uniform in our implementation). This procedure projects each operational token into a region of the embedding space that already corresponds to its intended semantics, rather than starting from random noise.

Intuitively, this initialization biases `<|backtracking|>` toward representations associated with control flow and reversal, while anchoring `<|OLD|>` and `<|NEW|>` near concepts of localization and repair.

Notablly, we observe this semantic anchoring stabilizes early-stage training. Without enabling semantic initialization, most models tuned on strict revision data would collapse into degenerate behaviors (e.g., never triggering revisions or always triggering at the start).

### D.2. Applying Alignment Algorithm

In addition to standard SFT, we explore the impact of different alignment algorithms on Stream of Revision's performance. In addition to the default SFT, we additionally experiment with DPO and SimPO, two prevalent offline preference optimization methods. We empirically observe that DPO and SimPO yield unstable revision triggering behavior, i.e., conducting excessive revisions (triggering 3.12x and 2.74x revisions on Qwen-2.5-7B compared to the SFT one), with similar security and utility performance. This indicates that SFT is more stable for this task when the data is limited.

## E. Training Details

SFT Training is conducted with DeepSpeed ZeRO-3 for memory efficiency. Maximum sequence length is 4,096 tokens. Optimization uses AdamW with a learning rate of 3e_-5, weight decay of 0.01, cosine scheduling, and a warmup ratio of 0.1. The per-device batch size is 2 with gradient accumulation of 2. Models are trained for 2 epochs in bfloat16 precision. We use a single NVIDIA A100 80GB GPU for all SFT runs.

## F. Alignment Data Construction Details

### F.1. Source Aggregation and De duplication

We aggregate CVE patch candidates from multiple public vulnerability databases and project advisories, then map each record to concrete version control commits. We deduplicate at two levels. First, we remove duplicates by commit hash. Second, we remove near duplicates by diff signature computed from normalized added and removed lines. We keep only examples where the remediation diff can be mapped to function level edits.

### F.2. Formal Construction of a Trajectory

Given an affected function $f$, we build an augmented trajectory $\mathcal{T}$ that contains normal code generation segments and one or more revision episodes. Each revision episode is defined as

$$\mathcal{E} = \tau_{\text{trig}} \oplus \langle\text{scope}\rangle s \langle/\text{scope}\rangle \oplus \langle\text{patch}\rangle s' \langle/\text{patch}\rangle$$

where $s$ is extracted as the contiguous span modified by a remediation hunk and $s'$ is the replacement span in the patched version.

### F.3. Bounded Trigger Latency

In real editing, developers may not trigger a revision immediately at the end of the vulnerable span. To reflect this behavior, we allow a bounded trigger latency when constructing training trajectories. Concretely, we permit up to $k$ normal code tokens after the end of $s$ before emitting $\tau_{\text{trig}}$. During data construction, we uniformly sample a trigger position within this window. In all experiments, we set $k$ to a fixed constant and keep it identical across models.

### F.4. Teacher Prompt for Security Neutral Specifications

For each function pair, we generate a specification $X$ using a teacher model with the following constraints. The teacher must describe only functional requirements that are common to both the vulnerable and patched versions. The teacher must not mention security, vulnerabilities, sanitization, bounds checking, memory safety, or any remediation related hints. We provide a small set of in context examples to enforce the neutrality requirement and manually verify a random subset for compliance.

### F.5. Strict and Relaxed Filtering Criteria

We define a code hunk as a contiguous diff region within a function. The strict set retains only commits where

1. exactly one function is modified

2. the modified function contains exactly one hunk

The relaxed set retains commits where

1. the number of modified functions is at most 5

2. each modified function contains at most 5 hunks

We use these two sets to study how supervision purity affects revision triggering frequency and inference token cost.

### F.6. General Instruction Replay

We mix revision trajectories with a general code instruction dataset to preserve coding utility and calibrate trigger behavior. We apply two filters to the general corpus. First, we retain samples that contain C or C++ code blocks. Second, we remove samples that explicitly discuss vulnerabilities or security patching to avoid leaking cues. We use a fixed mixing ratio $\lambda$ across all training runs.

## G. Agentic Baseline Construction

There is currently no off-the-shelf coding agent explicitly designed for secure code generation with a unified pipeline for vulnerability detection, localization, and repair. We initially experimented with general-purpose coding agents, including OpenHands and SWE-Agent. However, we found that these systems are unstable when paired with small-sized models we used in this paper (e.g., Qwen-2.5-7B). Common failure modes include tool invocation errors, uncontrolled output formats, and generation collapse across multi-turn interactions, leading to near-zero performance. These issues make them unsuitable as reproducible baselines for our study.

We therefore construct two controlled agentic baselines following the design principles of Agentless and common security patching workflows. Both baselines operate in a sequential multi-turn manner and rely solely on the underlying LLM without external static analyzers. They differ in whether vulnerability localization is explicitly separated from repair.

### G.1. Three-Step Agent (Generate–Detect–Repair)

This baseline executes three LLM calls:

1. **Generation:** produce a candidate solution given the task description.

2. **Detection:** audit the generated code for security vulnerabilities.

3. **Repair:** if vulnerabilities are detected, regenerate a secure version of the code.

The detection and repair prompts are defined as follows.

**Detection Prompt (Step 2).**

```
You are a security auditor.

Task: determine whether the following code contains security vulnerabilities.

Output requirements:
- Output MUST be a single JSON object and nothing else.
- Do NOT include Markdown fences.

Required JSON keys:
- is_vulnerable: boolean
- vulnerability_types: array of strings (e.g., CWE identifiers or short names)
- confidence: number between 0 and 1
- rationale: string (brief)

code:
{{CODE}}
```

**Repair Prompt (Step 3).**

```
You are a security code fixer.

Given code that may contain security vulnerabilities, output a patched version.

Output requirements:
- Output MUST be a single JSON object and nothing else.
- Do NOT include Markdown fences.

Required JSON keys:
- fixed_code: string (full code)
- changes: string (brief summary)

code:
{{CODE}}
```

This baseline resembles coarse-grained post-hoc repair, where the model re-encodes the entire program and produces a full rewritten version when vulnerabilities are detected.

### G.2. Four-Step Agent (Generate–Detect–Localize–Repair)

To reduce unnecessary regeneration and better reflect state-of-the-art agentic workflows, we construct a second baseline that explicitly separates localization from repair. This baseline executes four LLM calls:

1. **Generation:** produce a candidate solution.

2. **Detection:** determine whether vulnerabilities exist.

3. **Localization:** identify vulnerable regions in the code.

4. **Repair:** patch the code using both detection and localization results.

The prompts are defined as follows.

**Detection Prompt (Step 2).**

```
You are a security auditor.

Task: determine whether the following code contains security vulnerabilities.

Output requirements:
- Output MUST be a single JSON object and nothing else.
- Do NOT include Markdown fences.

Required JSON keys:
- is_vulnerable: boolean
- vulnerability_types: array of strings
- confidence: number between 0 and 1
- rationale: string (brief)

code:
{{CODE}}
```

**Localization Prompt (Step 3).**

```
You are a security auditor.

Task: locate the vulnerability locations in the given code.

Output requirements:
- Output MUST be a single JSON object and nothing else.
- Do NOT include Markdown fences.

Required JSON keys:
- locations: array of objects, each with:
    - function: string or null
    - description: string
    - snippet: string (a short code excerpt)

code:
{{CODE}}
```

**Repair Prompt (Step 4).**

```
You are a security code fixer.

Using the detection result and vulnerability locations, patch the code.

Output requirements:
- Output MUST be a single JSON object and nothing else.
- Do NOT include Markdown fences.

Required JSON keys:
- fixed_code: string (full code)
- changes: string (brief summary)

Detection JSON:
{{DETECTION_JSON}}

Localization JSON:
{{LOCALIZATION_JSON}}

code:
{{CODE}}
```

## H. Per-CWE Breakdown of Results

Figure 4 presents a per-category breakdown on the Top-10 CWEs from the 2025 Top-25 list. These CWEs correspond to the most common and impactful vulnerability classes in production software. We observe that Stream of Revision improves security or share on-par performance on 9 out of 10 categories. This fine-grained analysis supports our central claim: Stream of Revision does not merely shift the overall distribution, but enables targeted, just-in-time correction of high-risk local vulnerability patterns during generation. CWE-352 (Cross-Site Request Forgery) improves the most (safe one from 66 to 95) because CSRF vulnerabilities are typically caused by a localized missing guard (for example, absent token or origin validation), making them easy to detect during generation and fix with a small just in time revision.

## I. Case Study Examples

In this section, we present detailed case studies illustrating how Stream of Revision effectively identifies and rectifies security vulnerabilities during code generation, as shown in Figure 5. The function constructs a string by concatenating a message

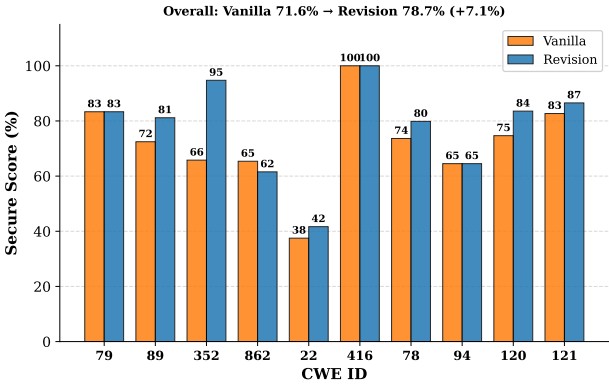

*Figure 4.* **Per-Category Secure Patch Rate (SPR) on Top-10 CWEs.** Comparison between the vanilla base model and Stream of Revision.

```
char *message_to_string(message_t *m) {
    [code prefix]
    ...
    char *buf = malloc(256);
    strcpy(buf, m->header);
    strcat(buf, ": ");
    strcat(buf, m->data);
    return buf;
<|backtracking|>
<|OLD|>
    char *buf = malloc(256);
    strcpy(buf, m->header);
    strcat(buf, ": ");
    strcat(buf, m->data);
    return buf;
<|/OLD|>
<|NEW|>
    char *buf = malloc(strlen(m->header) + strlen(m->data) + 3);
    strcpy(buf, m->header);
    strcat(buf, ": ");
    strcat(buf, m->data);
    return buf;
<|/NEW|>
}
```

*Figure 5.* **Case Study of Stream of Revision in Action.**

header, a delimiter, and the message body into a heap buffer. In the initial draft, the code allocates a fixed size buffer of 256 bytes and then uses unbounded string copy and concatenation routines to append the header and body. When either field is longer than the remaining capacity, this pattern can overflow the heap buffer, which is a classic memory corruption issue.

Stream of Revision triggers a revision and localizes the edit to the allocation statement while keeping the remaining logic unchanged. Specifically, it replaces the fixed allocation with a length aware allocation computed from the actual input sizes: the new buffer size is `strlen(header) + strlen(data) + 3`, where the constant accounts for the delimiter characters and the terminating null byte. After this change, the subsequent copy and concatenation operations fit within the allocated memory for typical null terminated inputs.

This revision directly removes the mismatch between a fixed capacity buffer and variable length inputs, which is the root cause of the overflow. The edit is small and localized, so it preserves the original formatting behavior while addressing the safety issue, which matches the intended design goal of making minimal, targeted corrections during generation.

While the revision mitigates the most immediate overflow risk, it does not fully harden the implementation. For example, it still assumes null terminated strings, does not check the return value of `malloc`, and it does not guard against integer overflow in the length computation.

## I.1. Analysis on Failure Cases

We first manually identify revisions that regress compilation from different runs. We randomly sample 20 cases for in-depth analysis. We find that 13 out of 20 cases are due to the model generating incomplete code snippets during the repair phase. While our localization mechanism correctly identifies the vulnerable span, we have no guarantee that the model will produce a syntactically valid replacement. Other cases are due to limitations of AST-based parsing that leads false positives. Our pipeline relies on syntax-level validation through AST parsing to ensure that rendered programs remain well-formed. However, in practice, existing AST parsers (e.g., tree-sitter and language-specific frontends) exhibit systematic limitations that inevitably introduce false positives. These issues are orthogonal to the proposed Stream of Revision mechanism and stem from the imperfect coverage and permissiveness of real-world parsers.

