# OpenReview forum: "Autoregressive, Yet Revisable: In Decoding Revision for Secure Code Generation"
_ICML.cc/2026/Conference — ICML 2026 regular_

### Official Review · Reviewer_towp · 2026-03-04

**Soundness:** 2
**Presentation:** 3
**Significance:** 2
**Originality:** 2
**Overall Recommendation:** 2
**Confidence:** 4

**Summary:**

The paper proposes Stream of Revision, a framework that enables LLMs to revise previously generated code during a single autoregressive decoding pass. Traditional code generation treats decoding as a strictly append-only process, which prevents models from correcting earlier mistakes and can propagate vulnerabilities once insecure code is produced. To address this limitation, the authors introduce special revision action tokens that allow the model to trigger a revision episode, localize a vulnerable span within the generated prefix, and synthesize a patch that replaces the problematic code. A deterministic renderer then applies these edits to produce the final program while keeping the generation stream autoregressive. The method is trained using revision trajectories constructed from real-world CVE vulnerability patches and is evaluated on secure code generation benchmarks such as CyberSecEval 2. Experiments show that the approach improves the security of generated code while preserving functional correctness and reducing computational overhead compared to multi-step agentic repair pipelines.

**Compliance With Llm Reviewing Policy:**

Affirmed.

**Final Justification:**

The main results presented in the paper are based on an unreliable benchmark due to limited context and inaccurate security labeling. This raises significant concerns about the soundness of the paper’s conclusions and contributions. The authors even themselves acknowledge such unreliability in the rebuttal.

The paper primarly targets vulnerability classes that are widely known and well-defined. Given this, the improvement from the paper (assuming it is correct) is marginal and in practice is unlikely to meaningfully reduce the burden on users, who must still assume that generated code may be insecure.

**Key Questions For Authors:**

1. Does CyberSecEval 2 consider sufficient code context when evaluating security?
2. Can the proposed approach handle larger, cross-function, or even cross-file fixes? If so, what would be the effectiveness and computation cost?

**Limitations:**

Please take a look at the weaknesses pointed out in the review, and either address them in the rebuttal or incorporate them as limitations of the paper.

**Strengths And Weaknesses:**

Strengths:

1. **Clear and intuitive problem framing.** The paper highlights a real limitation of standard autoregressive decoding for code generation, i.e., its inability to revise earlier tokens, which can lead to cascading errors and security vulnerabilities.

2. **Conceptually interesting decoding formulation.** The proposed Stream of Revision mechanism introduces explicit revision actions (trigger, localization, patching) within the decoding stream, which is a neat abstraction for enabling self-correction during generation.

3. **Lightweight approach.** The approach keeps the model fully autoregressive and uses a deterministic renderer to apply edits, avoiding multi-step agent pipelines and maintaining relatively low inference overhead.

Weaknesses:


1. **Limited novelty relative to existing approaches.** The idea of enabling editing during generation is related to prior work on constrained decoding and backtracking [1, 2]. However, the paper’s distinction from these approaches is somewhat incremental and not fully clarified.

2. **Poor benchmark choice.** The paper uses very specific benchmarks and the choice might be inappropriate. First, functional correctness is only evaluated on HumanEval, which is largely considered contaminated and saturated nowadays. Second, the security evaluation focuses on C/C++, for which vulnerabilities are known to be highly context-dependent [3]. The training set used in the paper is function level and ignores the necessary context, which may mislead the model. Moreover, the CyberSecEval 2 dataset may suffer from the same problem. It also relies on static checks for security, which might be inaccurate. These inappropriate benchmark choices pose a threat for the validity of the paper.

3. The security improvement compared to baselines is very modest (e.g., in Table 2). For security, the improvement of a few percent is not sufficient, as a large fraction of code still remains insecure and poses a threat for the user.

4. Computational trade-offs are not fully characterized. While the paper claims efficiency advantages over agentic approaches, it is potentially due to that the considered fixes are very local. The proposed approach may not be able to handle larger fixes, and even if so, might be expensive.

5. Most experiments use smaller open models, and it is unclear how well the approach scales to larger frontier models commonly used in practice.

[1] Ugare et al. "Itergen: Iterative semantic-aware structured LLM generation with backtracking." ICLR 2025.

[2] Mündler et al. "Type-constrained code generation with language models." PLDI 2025.

[3] Risse et al. "Top score on the wrong exam: On benchmarking in machine learning for vulnerability detection." ISSTA 2025.

---

> ### Author Rebuttal · Authors · 2026-03-31
>
> We thank the reviewer for the thoughtful and constructive feedback.
> > W1: Limited novelty relative to existing approaches. The idea of enabling editing during generation is related to prior work on constrained decoding and backtracking [1, 2].
>
> Our proposed approach SoR and [1,2] are different in four ways:
>
> * [1, 2] address formally specifiable properties (grammar rules and type systems), whereas SoR targets semantic vulnerabilities that no existing formal checker can express during partial code generation.
> * [1, 2] rely on external checkers to decide when to intervene while SoR leverages model's internal knowledge to trigger intervention via a learned special token.
> * [1, 2] are positionally constrained: they roll back contiguous tokens from the generation frontier; however, SoR introduces localization action and allows patching any span at an arbitrary position in code prefix.
> * [1, 2] performs token-level rollback; SoR conduct span-level detect-localize-repair actions.
>
> > W2a: Functional correctness is only evaluated on HumanEval, which is contaminated and saturated nowadays.
>
> To address this concern, we additionally evaluate SoR-tuned Qwen2.5-7B-Instruct on MBPP+ (Liu et al., EvalPlus) and LiveCodeBench-v5 (Jain et al.) in terms of pass@1, the latter of which is released after the training of base model and free from contamination.
>
> | Model| MBPP+| LiveCodeBench-v5|
> | - | - | - |
> | Qwen2.5-7B-Instruct | 0.683| 17.2|
> | SoR-tuned| 0.680| 17.0|
>
> The negligible difference confirms that SoR training preserves general functional correctness.
>
> > W2b: Second, the security evaluation focuses on C/C++, for which vulnerabilities are known to be highly context-dependent.
>
> We respectfully correct this claim: Table 2 reports Security Pass Rate across 6 languages (C, C++, C#, Java, JavaScript, Python), not C/C++ alone. While SoR trains only on C/C++ data, it shows consistent improvements across all evaluated languages.
>
> > W2c & Q1: Does CyberSecEval 2 consider sufficient code context when evaluating security? It also relies on static checks for security, which might be inaccurate.
>
> We directly address each concern. Regarding code context: our evaluation uses CSE2's instruct setting, where task instructions are synthesized from code snippets including context surrounding the insecure practice, so security-relevant context is preserved in the prompt. CSE2's own validation reports 96% precision and 79% recall for its static analyzer, which is imperfect but reasonable.
>
> To further validate under execution-based evaluation, we additionally evaluate CWEval (Peng et al., 2025), which uses dynamic test oracles instead of static pattern matching and provides expert-verified, self-contained instructions:
>
> | Language | Model| Functional (%) | Secure (%) | Func & Secure (%) |
> | - | -| -| -| -|
> |C| Qwen2.5-7B-Instruct | 35| 15| 10|
> | C| SoR-tuned| 40| 20| 15|
> | C++| Qwen2.5-7B-Instruct | 47| 33| 23|
> | C++| SoR-tuned| 47| 43| 29|
>
> SoR consistently improves security and remains functionality under dynamic evaluation, confirming that gains on CyberSecEval are not artifacts of static checker imprecision.
>
> > W3: The security improvement compared to baselines is very modest. For security, the improvement of a few percent is not sufficient, as a large fraction of code still remains insecure and poses a threat to the user.
>
> We respectfully note that secure code generation remains an open challenge, in analogous settings like SWE-bench, improvements of ~5% are considered significant. SoR achieves 8–15% improvement over vanilla models with only 1.06× token overhead and no external tools (Table 3), which we consider a non-trivial gain.
>
> Moreover, SoR is orthogonal to existing weight-level alignment methods. Table 4 confirms that SoR + ProSec outperforms either alone (C++: 79.75→83.90), demonstrating its value as a complementary defense layer.
>
> > W4 & Q2: Can the proposed approach handle larger, cross-function, or even cross-file fixes? If so, what would be the effectiveness and computation cost?
>
> SoR is designed for local revisions by construction. This scope is deliberate, as over half of real-world CVE patches involve single-hunk modifications within a function (Appendix A). We position SoR as the first step towards in-decoding intervention for secure code generation. It naturally applies to cross-function/file generation scenarios: because it intervenes at each decoding step, local vulnerabilities are detected and repaired as each function is generated, with the same 1.06× overhead (Sec 3.4).
>
> SoR's architecture also supports multi-span revision by chaining multiple localization-and-repair episodes. However, multi-hunk commits in existing datasets often bundle non-security modifications (e.g., refactoring) that introduce label noise (Sec 4.2). We therefore restrict training to single-hunk patches to ensure data quality; developing better filtering strategies for multi-hunk training is a concrete future direction.

---

> > ### Author Rebuttal · Reviewer_towp · 2026-04-04
> >
> > Thank you for the detailed rebuttal. While several points provide helpful clarification, others introduce new major concerns. Please see below.
> >
> > > W1: [1, 2] address formally specifiable properties (grammar rules and type systems), whereas SoR targets semantic vulnerabilities that no existing formal checker can express during partial code generation.
> >
> > This distinction does not seem entirely accurate. Many of the vulnerabilities studied in the paper (e.g., those shown in Figure 4) can in fact be formally specified. Moreover, it is known that important classes of vulnerabilities, such as memory safety issues in C/C++, can be systematically prevented in languages like safe Rust through strong type systems. Therefore, the boundary between “formally specifiable properties” and the vulnerabilities targeted by SoR appears less clear-cut than suggested.
> >
> > > W2c & Q1: Regarding code context: our evaluation uses CSE2's instruct setting, where task instructions are synthesized from code snippets including context surrounding the insecure practice, so security-relevant context is preserved in the prompt.
> >
> > This explanation raises concerns. Accurately identifying and extracting security-relevant context is itself a well-known challenge. As such, either (i) the extracted context may be incomplete or imprecise, or (ii) the evaluation is limited to relatively simple scenarios where context is easy to recover. In either case, this weakens confidence in how well the setup reflects realistic code generation settings.
> >
> > > W2c & Q1: CSE2's own validation reports 96% precision and 79% recall for its static analyzer, which is imperfect but reasonable.
> >
> > These numbers appear relatively low, particularly the recall. Given that the paper’s evaluation heavily depends on CSE2, this level of imperfection could significantly affect the validity of the reported results. In particular, missed vulnerabilities (false negatives) may lead to overly optimistic conclusions about security performance.
> >
> > > W3: in analogous settings like SWE-bench
> >
> > The comparison to SWE-bench does not appear well justified. SWE-bench evaluates functional correctness from informal natural language intent, where some degree of imperfection is expected and strong reasoning is required to complete tasks. In contrast, secure code generation, at least for many vulnerability classes considered in this paper, often involves well-defined, formally specifiable properties. Such properties can, in principle, be enforced to a much higher (potentially near-perfect) standard. As a result, the analogy is potentially misleading, and the reported improvements seem modest in this context.

---

> > > ### Author Response · Authors · 2026-04-08
> > >
> > > We thank the reviewer for the continued engagement and for acknowledging that several points in our rebuttal provide helpful clarification.
> > >
> > > In particular, we are glad that our new results on MBPP+ and LiveCodeBench (W2a), the clarification that Table 2 covers 6 languages rather than C/C++ alone (W2b), and our discussion of cross-function applicability as well as scalability (W4, W5) appear to have resolved those concerns. Regarding W1 (novelty), we note that three of our four stated distinctions from existing approaches also appear to be accepted.
> > >
> > > We address the remaining points below.
> > > > W1a — On the distinction from [1, 2]: The reviewer highlights that (i) many vulnerabilities in our paper can be formally specified and addressed by formal checkers, and (ii) memory safety issues can be systematically prevented through strong type systems in languages like safe Rust.
> > >
> > > We appreciate the reviewer's nuanced point and agree that some vulnerability classes (e.g., memory safety) can be formally specified on complete programs in certain languages, safe Rust being an excellent example. However, our distinction holds on two levels.
> > > + First, many important vulnerability classes cannot or are hard to formally specify. For example, the Top-10 most dangerous CWEs include authorization flaws, path traversal, and CSRF, many of which depend on code semantics and intended program behavior, not structural properties that a formal checker can express. SoR improves on 9 out of 10 of these top CWEs (Figure 6) by leveraging models' security prior to dynamically localize and repair vulnerable code spans, extending well beyond the formally specified subset.
> > > + Regarding the safe Rust example: we agree that strong type systems can prevent memory safety issues. However, Rust's type system addresses memory safety but not injection, CSRF, authorization, or many other vulnerability classes that SoR targets. Again, the 9/10 top-CWE coverage demonstrates that SoR addresses a considerably broader scope than any single PL-level mitigation. Moreover, switching programming languages is outside the scope of LLM-based code generation; C/C++ remains necessary in many real-world settings (e.g., systems programming, embedded systems).
> > >
> > > > W2c & Q1 — On CSE2's context extraction and static analysis reliability. The reviewer raises two concerns about CSE2: (i) that automated extraction of security-relevant context may be incomplete or limit evaluation to simple scenarios, and (ii) that 79% recall of the static analyzer used by CSE2 may lead to overly optimistic security conclusions.
> > >
> > > We share the reviewer's intuition that no single benchmark's setup is perfect. This is precisely why we provided CWEval results (Peng et al., 2025) in our rebuttal, a benchmark that uses expert-verified, self-contained task instructions and dynamic test oracles, entirely bypassing automated context extraction and static analysis.
> > >
> > > | Language| Model| Functional (%) | Secure (%) | Func & Secure (%) |
> > > |-|-|-|-|-|
> > > | C| Qwen2.5-7B-Instruct| 35|15|10|
> > > | C| SoR-tuned| 40| 20|15|
> > > | C++| Qwen2.5-7B-Instruct| 47|33|23|
> > > | C++| SoR-tuned|47|43|29|
> > >
> > > SoR shows consistent improvements under CWEval, demonstrating that the gains are not artifacts of CSE2's context setup. We view the convergence of two independent evaluation methodologies, i.e., CSE2 and CWEval, as substantive evidence that our findings are robust.
> > >
> > > > W3 — On improvement magnitude. The reported improvements seem modest in this context.
> > >
> > > To contextualize the reported improvements, we note that SoR achieves 8–15% improvement with only 1.06× overhead and no external tools, and is composable with existing alignment methods (Table 4: SoR + ProSec, C++ 79.75→83.90).
> > >
> > > To put this in perspective, we conducted additional experiments integrating Semgrep — a widely-used static analyzer that enforces precisely the kind of formally specifiable security properties the reviewer describes — into an agentic repair pipeline by replacing the LLM-based vulnerability detection stage with SAST justification. However, we observe that, despite doubling inference cost and leveraging external security signals, this integration does not consistently improve performance and underperforms SoR.
> > >
> > > | Method|C|C++|
> > > |-|-|-|
> > > | Post-hoc Agent w/o loc|67.86|79.75|
> > > | Post-hoc Agent w/ loc|70.87|82.23|
> > > | Post-hoc Agent w/o loc + Semgrep|65.35|81.82|
> > > | Post-hoc Agent w/ loc + Semgrep|66.34|82.64|
> > > | SoR (Ours)|**73.76**|**83.47**|
> > >
> > > We also respectfully note that the expectation of "near-perfect" enforcement does not reflect the current state of the art. As discussed in W1, many vulnerability classes depend on code semantics that no existing formal checker can express. More broadly, Table 2 and the above experiment confirm that no method, even with external tool support, achieves near-perfect secure code generation. Secure code generation remains an open scientific challenge, and we believe SoR makes a meaningful step toward this long-term goal.

---

### Official Review · Reviewer_6hox · 2026-03-08

**Soundness:** 4
**Presentation:** 3
**Significance:** 3
**Originality:** 3
**Overall Recommendation:** 4
**Confidence:** 4

**Summary:**

This paper makes revision an internal action of decoding by introducing explicit trigger / scope / patch tokens, allowing the model to revise previously generated prefixes during generation. The method is trained on CVE patch pairs for secure code generation, and the tables directly support both its target-domain security gains and its token-efficiency advantage. The main caveats are that the method explicitly relies on a localized vulnerability assumption, and OOD utility is not evaluated as thoroughly as OOD security.

**Compliance With Llm Reviewing Policy:**

Affirmed.

**Final Justification:**

My concerns have been resolved, so I would like to increase soundness score to 4, and keep my overall score of 4.

**Key Questions For Authors:**

- How strong is the method’s dependence on the locality assumption? In particular, how many failure cases are not well modeled as a single localized patch?
- The post-hoc agent baseline is constructed by the authors and explicitly does not use external static analyzers. How would the comparison look against stronger repair systems with static-analysis or verifier-based support?
- CyberSecEval 2 is a reasonable benchmark, but it is still not the same as real exploit success rate. So the paper currently shows that the method is more likely to pass benchmark security checks, not that it truly eliminates vulnerability risk. Can the authors clarify how they view this gap, and whether they have additional evidence beyond benchmark pass rates?

**Limitations:**

As discussed above

**Strengths And Weaknesses:**

Strengths
- The paper addresses a structural limitation of standard autoregressive decoding, rather than simply adding another post-hoc repair stage after generation.
- The scope localization mechanism is well designed. In particular, substring-constrained scope generation is a sensible way to avoid hallucinated edit locations.
- The empirical evaluation is strong overall. The method improves security across multiple backbones, shows some zero-shot transfer beyond the training languages, and has a clear efficiency advantage over the post-hoc repair baseline.
- The paper also considers utility preservation rather than optimizing only for security, which is important for this problem setting.

Weaknesses
- The method relies on a fairly strong localized vulnerability assumption and is best suited to local, contiguous, patch-like fixes. The paper does not provide enough support for more global or interdependent code revisions.
- Some of the broader generalization claims feel overstated. The results do suggest useful transfer, but they do not yet strongly establish language-agnostic security reasoning.

---

> ### Author Rebuttal · Authors · 2026-03-31
>
> We thank the reviewer for the thoughtful and constructive feedback.
> > **Q1:** How strong is the method’s dependence on the locality assumption? In particular, how many failure cases are not well modeled as a single localized patch?
>
> SoR is designed for local revision by construction. This scope is deliberate and covers a substantial portion of real-world vulnerabilities: over half of CVE patches in commonly used datasets (BigVul, PrimeVul, CVEFixes) involve single-hunk modifications within a function. For cases outside this scope, in our manual inspection, SoR rarely triggers revision, leaving the base model's generation unchanged with no additional failure risk. We position SoR as the first step towards in-decoding intervention for secure code generation.
>
> To further reduce these out-of-scope cases, SoR architecturally supports extension to multi-span revision by chaining multiple localization-and-repair episodes. However, as discussed in Sec 4.2, real-world vulnerability datasets are constructed from vulnerability-fixing commits, and multi-hunk commits often contain non-security modifications like code refactoring that introduce noise into training data. As a proof-of-concept, we therefore deliberately restrict training to single-hunk patches to ensure data quality. Developing better data filtering strategies for multi-hunk training is a concrete future direction.
>
> > **Q2:** The post-hoc agent baseline is constructed by the authors and does not use external static analyzers. How would the comparison look against stronger repair systems with static-analysis or verifier-based support?
>
> We thank the reviewer for this suggestion. We conducted additional experiments by integrating Semgrep (a widely-used static analyzer [1]) into the agentic baseline, replacing the LLM-based classification stage. Only code flagged as vulnerable by Semgrep is sent to the downstream repair pipeline.
>
> | Method | C | C++ |
> | --- | --- | --- |
> | Post-hoc Agent w/o localization | 67.86 | 79.75 |
> | Post-hoc Agent w/ localization | 70.87 | 82.23 |
> | Post-hoc Agent w/o loc + Semgrep | 65.35 | 81.82 |
> | Post-hoc Agent w/ loc + Semgrep | 66.34 | 82.64 |
> | Stream of Revision (Ours) | **73.76** | **83.47** |
>
> Results show that SoR continues to outperform all agent variants (C: 73.76, C++: 83.47). Meanwhile, integrating Semgrep does not consistently improve agent performance: on C, pass rates decrease (w/o loc: 67.86→65.35; w/ loc: 70.87→66.34), while C++ shows marginal gains (w/o loc: 79.75→81.82; w/ loc: 82.23→82.64). We attribute this to Semgrep's own false positive/negative rates, introducing noise into downstream repair.
>
> **We highlight that this comparison is inherently favorable to the agent baselines**: they leverage external security signals (Semgrep's rules) and require additional inference passes with significantly higher token cost, whereas SoR relies solely on internalized knowledge with only 1.06× token overhead in a single inference. Despite these asymmetries, SoR still outperforms all agent variants.
>
> [1] https://github.com/semgrep/semgrep
>
> > **Q3:** So the paper currently shows that the method is more likely to pass benchmark security checks, not that it truly eliminates vulnerability risk. Can the authors clarify how they view this gap, and whether they have additional evidence beyond benchmark pass rates?
>
> We appreciate reviewer's insightful suggestion.
>
> We acknowledge that benchmark pass rates are not equal to eliminating all vulnerability risk. This is a broadly recognized limitation in secure code generation evaluation, shared by all methods in Table 2.
>
> To provide evidence beyond static checks, we offer two additional analyses.
>
> First, two authors of this paper with security experience manually reviewed 20 randomly sampled SoR outputs on CSE2. Among 19 triggered revisions, 15 are correct vulnerability fix attempts. However, 3 cases of revised code still contained other vulnerabilities that CSE2's predefined patterns did not flag, confirming that static checkers verify known patterns but cannot guarantee vulnerability-free code. We emphasize that this limitation affects **all approaches** in Table 2 equally and does not significantly alter the relative improvements SoR achieves over the baselines.
>
> Second, we evaluate on CWEval [2], which uses outcome-driven dynamic test oracles instead of static analyzers, offering more rigorous evaluation than static pattern matching. SoR consistently improves security under this stricter setting:
>
> | Language | Model | Functional (%) | Secure (%) | Func & Secure (%) |
> | --- | --- | --- | --- | --- |
> | C | Qwen2.5-7B-Instruct | 35 | 15 | 10 |
> | C | SoR-tuned | 40 | 20 | 15 |
> | C++ | Qwen2.5-7B-Instruct | 47 | 33 | 23 |
> | C++ | SoR-tuned | 47 | 43 | 29 |
>
> SoR consistently improves security under dynamic evaluation, confirming that gains on CSE2 are not artifacts of static checker imprecision.
>
> [2] CWEval: Outcome-driven Evaluation on Functionality and Security of LLM Code Generation

---

> > ### Author Rebuttal · Reviewer_6hox · 2026-04-02
> >
> > I thank the authors for their detailed rebuttal. All of my concerns have been resolved, and I'd like to keep my score of 4.

---

### Official Review · Reviewer_xB6M · 2026-03-12

**Soundness:** 4
**Presentation:** 3
**Significance:** 4
**Originality:** 3
**Overall Recommendation:** 5
**Confidence:** 4

**Summary:**

This paper proposes Stream of Revision, which changes the code generation from a monotonic stream to a dynamic, self-correcting trajectory. So that the model can achieve jit self-correction. This paper use the method design for secure code generation.

**Compliance With Llm Reviewing Policy:**

Affirmed.

**Final Justification:**

I will maintain my original assessment.

**Key Questions For Authors:**

How to make the method general for secure code generation?

**Limitations:**

yes

**Strengths And Weaknesses:**

Pros:
1. Novel idea and a interesting design for revision episode.
2. The results are convincing on C/C++ and the efficiency is good.

Cons:
1. In failure analysis, many of failure samples are due to incompleted generation.
2. The local vulnerability makes the method limited to local and frequent issues.

---

> ### Author Rebuttal · Authors · 2026-03-31
>
> We thank the reviewer for the thoughtful and constructive feedback.
>
> > Q1 How to make the method general for secure code generation?
>
> We first note that SoR already demonstrates meaningful generality: trained exclusively on C/C++, it achieves zero-shot security improvements across Python, Java, C#, and JavaScript (Table 2), and improves 9/10 Top-10 CWEs (Appendix, Figure 6), suggesting it learns transferable security patterns.
>
> To further improve generality, we identify two directions.
>
> First, regarding incomplete generation (W1): this is the primary failure mode in our analysis, where the model produces incomplete patches during revision. We believe this is addressable by training with stronger base models with better long-form generation capability, and by augmenting training data with more diverse patch lengths.
>
> Second, regarding the locality scope (W2): SoR architecturally supports chaining multiple localization-and-repair episodes within one decoding pass. Specifically, we can transform real-world multi-hunk vulnerability patches into training trajectories with chained localization-and-repair episodes. However, as discussed in Sec 4.2, the current bottleneck is training data quality: real-world vulnerability datasets are constructed from vulnerability-fixing commits, in which multi-hunk commits often contain non-security modifications, such as code refactors, that introduce noise into the training data. Developing better data filtering strategies for multi-hunk training is our concrete future direction.

---

> > ### Author Rebuttal · Reviewer_xB6M · 2026-04-03
> >
> > Thank you for the rebuttal. The zero-shot transfer results and the concrete future directions for addressing incomplete generation and multi-hunk vulnerabilities adequately address my concerns.

---

### Official Review · Reviewer_s6qs · 2026-03-13

**Soundness:** 4
**Presentation:** 4
**Significance:** 4
**Originality:** 4
**Overall Recommendation:** 4
**Confidence:** 4

**Summary:**

The paper explores a significant challenge in LLM-based code generation:
LLMs are autoregressive, which makes it hard to accomplish iterative and revisable programming like humans revise via in-place rewriting.
Failing to do so leads to error propagation and a "commitment trap" where early defects cannot be easily corrected.

To address this, the paper proposes *Stream of Revision*, a novel paradigm that internalizes the revision process directly into the LLM's single forward pass. This is achieved by augmenting the model's vocabulary with special "action tokens" that enable dynamic backtracking and in-place editing. Specifically, revision is structured into three stages: a revision trigger token for vulnerability detection, content-addressable localization of the vulnerable code span using a "Strict Substring Invariant," and explicit patch synthesis. A deterministic renderer then applies these revision actions to produce the final, corrected program.

The authors validate the efficacy of Stream of Revision on secure code generation.

**Compliance With Llm Reviewing Policy:**

Affirmed.

**Final Justification:**

The authors' rebuttal has addressed my main concerns. I maintain a positive assessment and have raised confidence to 4.

**Key Questions For Authors:**

1. **Q1: Security Depth in Case Study.**

   The case study (Figure 5) is illustrative but also highlights that the revision "does not fully harden the implementation" (e.g., `malloc` return value check, integer overflow in length computation).
   While being honest about limitations is good, it raises a question about the 'depth' of security fixes the model learns.
   Is it primarily learning to fix obvious, direct vulnerabilities, or can it be steered towards more comprehensive secure coding practices?

2. **Q2: Global/local revisions.**

   Could the authors elaborate on the limitations of their current approach for more global code revisions or "asynchronous edits across multiple hunks", as mentioned in Appendix A? How might the Stream of Revision paradigm be extended to handle such cases, or is it fundamentally designed for localized, just-in-time corrections? A response here would clarify the scope and potential future directions for addressing more complex vulnerability types.

3. **Q3: Non-security-critical revisions.**

   The qualitative observation notes that one-third of revisions are for "code refactoring" rather than security, introducing "additional yet light computational overhead". Could the authors discuss strategies to either reduce these non-security-related revisions (e.g., through modified training objectives, negative sampling, or a separate "intent" token) or to differentiate their computational cost from security-critical revisions? Clarifying this would help in optimizing the model's behavior and efficiency.

4. **Q4: Security level of code revisions.**

   The case study (Figure 5) shows that even after revision, some potential issues remain (e.g., `malloc` return value check, integer overflow). Is the current goal of "Stream of Revision" primarily to address the most prominent/direct vulnerabilities to reach a "good enough" security state, or do the authors envision pathways for it to learn and apply more comprehensive hardening strategies, perhaps through multi-turn revisions or different training signals? This would clarify the ultimate ambition for the model's security capabilities.

5. **Q5: Figure annotation.**

   Pass rate annotations of Figure 3 (SPR: Relaxed Samples) are partially covered by neighboring bars.

**Limitations:**

Yes.

**Strengths And Weaknesses:**

### Strengths

1. **Soundness:**
   The method design, experiment design and result analysis are generally sound.
2. **Presentation:**
   The paper is well-written and easy to follow. Figures and tables are well presented.
3. **Significance:**
   This work address the interesting and timely topic of secure code generation.
   Most importantly, it highlights a paradigm shift, which makes LLMs capable of revising and correcting past code.
   Future work could benefit from this idea, especially when the submission has provided its code in the anonymous repository.
   Moreover, Stream of Revision has the potential to be of practical value in real-world LLM-assisted code generation applications.
4. **Originality:**
   The methodology of this paper is novel and original.

### Weaknesses

1. **W1: Scope of Revision.**

   While I appreciate that the paper explicitly acknowledges the limitation in Appendix A regarding "asynchronous edits across multiple hunks", the core efficiency argument ($\mathcal{O}(1)$ overhead) relies heavily on the assumption of localized vulnerabilities. It would be beneficial to discuss how the system would behave or be extended for more complex, global refactoring tasks or vulnerabilities requiring non-contiguous or very large-scale changes. While outside the current scope, a brief thought on future directions for such scenarios would strengthen the paper.
2. **W2: Qualitative Analysis of Revision Types.**

   The paper notes that one-third of revisions are for "code refactoring" rather than security, which introduces "additional yet light computational overhead." While acknowledged, it would be valuable to explore further if these non-security revisions are always beneficial or sometimes undesirable. Discussion on potential strategies to explicitly bias the model towards security-critical revisions or to differentiate between beneficial refactoring and unnecessary churn would be insightful.

---

> ### Author Rebuttal · Authors · 2026-03-31
>
> We thank the reviewer for the thoughtful and constructive feedback.
>
> > **Q1:** While being honest about limitations is good, it raises a question about the 'depth' of security fixes the model learns. Is it primarily learning to fix obvious, direct vulnerabilities, or can it be steered towards more comprehensive secure coding practices?
>
> SoR is able to learn non-trivial security fixes. Our per-CWE analysis (Appendix, Figure 6) shows improvements on 9/10 Top-10 most-dangerous CWEs, spanning diverse and complex vulnerability classes, including buffer overflow, injection, CSRF, and improper access control.
>
> More broadly, revision depth heavily depends on the base model's prior knowledge of security, where SoR-training actually aims to activate the model's existing security knowledge into actionable code revisions. Stronger base models would naturally produce deeper fixes through the same framework via stronger security prior.
>
> > **Q2:** It would be beneficial to discuss how the system would behave or be extended for more complex, global refactoring tasks or vulnerabilities requiring non-contiguous or very large-scale changes. While outside the current scope, a brief thought on future directions for such scenarios would strengthen the paper.
>
> We thank the reviewer for this suggestion and will include a dedicated discussion of these directions in the revision. SoR architecturally supports extension to multi-hunk revisions by chaining multiple localization-and-repair episodes after triggering the revision action within one decoding pass. However, as discussed in Sec 4.2, the current bottleneck is training data quality: real-world vulnerability datasets are constructed from vulnerability-fixing commits, and multi-hunk commits often contain non-security modifications like code refactoring that introduce noise into training data. Developing better data filtering strategies for multi-hunk training is our concrete future direction.
>
> For cross-file/component generation, SoR operates normally within each generation step, handling local repairs with the same overhead guarantees. However, cross-location vulnerability fixes, where the root cause and patch span different components, require a global context that is not fully accessible within the model's single decoding context. Since SoR acts on the model context in the current decoding stream, these vulnerabilities are better addressed by post-hoc approaches with full program visibility. We view SoR as complementary to such methods, and exploring their integration is a promising future direction.
>
> > **Q3:** Could the authors discuss strategies to either reduce these non-security-related revisions (e.g., through modified training objectives, negative sampling, or a separate "intent" token) or to differentiate their computational cost from security-critical revisions?
>
> We find two directions may be promising for reducing non-security revisions and will discuss them in the revision.
>
> First, our logit calibration mechanism (Section 5.4) provides direct control over trigger frequency by adjusting the logit bias of the revision trigger token during inference. We observe that lower bias reduces refactoring revisions more than security-critical ones, as the model is more confident when triggering genuine vulnerabilities during training.
>
> Second, the reviewer's suggestion of negative sampling is valuable—for example, constructing negative samples containing only refactoring revisions and applying DPO to suppress non-security triggers while preserving security-critical ones. We plan to explore this direction in the future.
>
> > **Q4:** Is the current goal of "Stream of Revision" primarily to address the most prominent/direct vulnerabilities to reach a "good enough" security state, or do the authors envision pathways for it to learn and apply more comprehensive hardening strategies, perhaps through multi-turn revisions or different training signals?
>
> Our response to Q1 clarifies the current goal of SoR. Meanwhile, we do envision pathways towards more comprehensive hardening. Multi-turn revisions via chained localization-and-repair episodes are architecturally supported (as in our response to Q2), which could benefit fixing complex vulnerabilities within a single decoding process. We also believe richer training signals, such as synthesizing training trajectories following defensive coding practices beyond minimal CVE patches, could steer the model towards deeper fixes. Additionally, since SoR aims to activate the base model's security knowledge, stronger base models with richer security priors would naturally yield deeper fixes.
>
>
> > **Q5:** Pass rate annotations of Figure 3 (SPR: Relaxed Samples) are partially covered by neighboring bars.
>
> Thank you for pointing this out. We will fix the overlapping annotations in Figure 3 in the revised version.

---

> > ### Author Rebuttal · Reviewer_s6qs · 2026-04-01
> >
> > I thank the authors for their rebuttal, and most of my concerns have been addressed. I lean towards acceptance and will raise my confidence to 4.

---

### Decision · Program_Chairs · 2026-04-30

**Decision:**

Accept (regular)

**Comment:**

This paper introduces  an elegant paradigm that enables Large Language Models to self-correct vulnerabilities during a single autoregressive decoding pass. By augmenting the vocabulary with specific action tokens (trigger, localize, patch), the model can backtrack and edit its own history without relying on high-latency external tools or post-hoc agentic pipelines.

The reviewing committee appreciated the conceptual novelty and practical efficiency of internalizing the revision loop. The empirical evidence demonstrates that SoR reduces vulnerabilities across multiple programming languages with minimal inference overhead.

During the rebuttal, the authors rigorously addressed the primary concerns raised by the reviewers:

Evaluation Robustness: To counter valid concerns regarding the limitations of static analysis in CyberSecEval 2 and potential contamination in HumanEval, the authors provided strong additional results on uncontaminated functional benchmarks (MBPP+, LiveCodeBench-v5) and dynamic security test oracles (CWEval).

Baseline Comparisons: The authors demonstrated that SoR outperforms an agentic pipeline augmented with a static analyzer (Semgrep), highlighting the efficacy of the internalized approach.

While one reviewer strongly advocated for rejection—arguing that the security gains are modest and that such vulnerabilities are better solved via formal methods or memory-safe languages (e.g., Rust)—the consensus is that advancing secure C/C++ generation remains a critical and highly relevant challenge. An 8-15% security improvement with only a ~1.06x token overhead is a non-trivial algorithmic achievement for autoregressive models.

Required for Camera-Ready:
The authors shall incorporate the robust additions from their rebuttal into the final manuscript. This includes the MBPP+/LiveCodeBench-v5 and CWEval results, the Semgrep baseline comparison, and a clear discussion of the method's current limitations regarding multi-hunk/global revisions and non-security refactoring overhead.